# Detection of HEV RNA Using One-Step Real-Time RT-PCR in Farrow-to-Finish Pig Farms in Bulgaria

**DOI:** 10.3390/pathogens12050673

**Published:** 2023-05-03

**Authors:** Gergana Lyubomirova Krumova-Valcheva, Ilaria Di Bartolo, Richard Piers Smith, Eva Gyurova, Gergana Mateva, Mihail Milanov, Albena Dimitrova, Elke Burow, Hristo Daskalov

**Affiliations:** 1National Centre for Food Safety, National Diagnostic and Research Veterinary Medical Institute, 1606 Sofia, Bulgaria; dr.krumova_vulcheva@abv.bg (G.L.K.-V.); evaguirova@abv.bg (E.G.); gerymat@abv.bg (G.M.); m.vlad.milanov@gmail.com (M.M.); aldimas@abv.bg (A.D.); 2Departement of Food Safety, Nutrition and Veterinary Public Health, Istituto Superiore di Sanità, 00161 Roma, Italy; ilaria.dibartolo@iss.it; 3Animal and Plant Health Agency—Weybridge, Addlestone KT15 3NB, UK; richard.p.smith@apha.gov.uk; 4Department Biological Safety, Federal Institute for Risk Assessment, 12277 Berlin, Germany; elkeburow@hotmail.de

**Keywords:** hepatitis E virus, farrow-to-finish pig farms, finishers, dry sows, gilts, one-step real-time RT-PCR

## Abstract

(1) Background: HEV is a zoonotic, foodborne pathogen. It is spread worldwide and represents a public health risk. The aim of this study was to evaluate the presence of HEV RNA in farrow-to-finish pig farms in different regions of Bulgaria; (2) Methods: Isolation of HEV RNA from pooled samples of feces was performed using a QIAamp^®^ Viral RNA Mini Kit followed by HEV RNA detection using a single-step real-time RT-PCR with primers and probes targeting the ORF 3 HEV genome; (3) Results: HEV RNA was detected in 12 out of 32 tested farms in Bulgaria (37.5%). The overall percentage of HEV-positive pooled fecal samples was 10.8% (68 of 630 samples). HEV was detected mostly in pooled fecal samples from finisher pigs (66/320, 20.6%) and sporadically from dry sows (1/62, 1.6%) and gilts (1/248, 0.4%); (4) Conclusions: Our results confirm that HEV circulates in farrow-to-finish pig farms in Bulgaria. In our study, we found HEV RNA in pooled fecal samples from fattening pigs (4–6-months age), shortly before their transport to the slaughterhouse indicating a potential risk to public health. The possible circulation of HEV throughout pork production requires monitoring and containment measures.

## 1. Introduction

Hepatitis E is caused by the hepatitis E virus, a single-stranded RNA virus, belonging to the Family *Hepeviridae* [1], which was recently classified into two subfamilies [2]. The *Orthohepevirinae subfamily* infects, mammals, and birds, and includes the species *Paslahepevirus* which infects humans and several animal species, including zoonotic genotypes. Eight genotypes have been described in the species *Paslahepevirus balayani*, four of which are significant causative agents of diseases in humans. Genotypes 1 and 2 cause endemic waterborne outbreaks in developing countries, while genotypes 3 and 4 are zoonotic, causing sporadic cases and small foodborne outbreaks mainly in industrialized countries. Genotype 3 is widespread worldwide and infects humans, domestic pigs, wild boar, deer, and rabbits. It is also the most common genotype found in Europe [3].

Hepatitis E infection in people ranges from asymptomatic to self-limiting icteric hepatitis [4]. The incubation period is from 2 to 10 weeks, and infected people can excrete the virus for several days up to 2–3 weeks after clinical symptoms have been resolved [5]. However, in recent years, sufficient data have been collected to indicate that hepatitis E virus infection can be a serious threat [6,7,8]. In industrialized countries, HEV can cause severe or chronic liver disease in people with liver damage and in immunocompromised patients [9,10]. The disease can be fatal in pregnant women in low-income countries [11].

Hepatitis E infection is becoming increasingly relevant in European countries. From 2005 to 2015, more than 21,000 acute cases and more than 28 deaths have been reported [12]. Sporadic cases of hepatitis E have been reported in the United Kingdom, France, Italy, Spain, Netherlands, Greece, Germany, Austria, Poland, and Bulgaria [6,13,14]. This could be linked to higher awareness of clinicians or represent a real increase in hepatitis E spread in Europe, suggesting a change in the risk of virus transmission. 

Domestic pigs are the main reservoir of hepatitis E virus [12] and, therefore, the foodborne transmission of the pathogen should not be underestimated. Infected pigs and their related food products could, thus, present a risk to the people. Some studies have detected the presence of HEV in pigs at the slaughterhouses in their liver and feces [15,16,17] or in muscles and serum where it is rarely detected [18]. HEV has also been detected in pork products at the point of sale [12]. These studies confirm the wide presence of HEV RNA. An EFSA opinion, summarizing the potential for zoonotic transmission, identified the risk through the consumption of raw and undercooked pork products containing HEV [12]. 

Several studies also confirmed the widespread circulation of HEV-3 in wild boar that, along with domestic pigs, are the main reservoirs of this zoonotic genotype [19]. Some other hosts, such as deer and rabbits, are also a reservoir of HEV-3. Human cases have been linked to the consumption of raw or undercooked pig liver, deer sashimi, and wild boar [6,7,20,21]. In addition, some studies supported that moose, rats, and ferrets have also been reported to carry host-specific variants of HEV; although, at present there is no certain evidence of HEV transmission to humans [12]. Very recently, a few human cases were positive for rat-HEV, which belongs to another genus [22]. 

Some studies reported a higher HEV seroprevalence among workers in contact with pigs, such as veterinarians, pig farmers, and forestry officials [23], suggesting that direct contact with infected animals could be another route of transmission to humans.

Data from Bulgaria are limited. A serological study, conducted in 2021, among Bulgarian blood donors showed an overall seroprevalence of 25.9% for anti-HEV IgG [13], confirming circulation of the virus in the Bulgarian population [14,24,25]. In animals, a study conducted during 2017–2019 demonstrated the presence of anti-HEV IgG in 75.8% (91/120) of tested pig samples (5–6 month of age) [26]. Another study concerning fattening pigs in three industrial farms located in Northern Bulgaria also showed high (~40%) seroprevalence [27]. 

The aim of this study was to determine the presence of HEV RNA in industrial pig farms in different regions of Bulgaria. These regions have the highest pig farm density in the country. The study will also give us a first indication of the exposure of pigs to the virus in different age groups, up to the age before slaughter for human consumption. 

## 2. Materials and Methods

### 2.1. Sampling

In the context of the One Health European Joint Program project (№ 773830) “Biosecurity practice for pig farming across Europe (BIOPIGEE)”, the presence of HEV in pig farms was assessed in 9 countries. Farrow-to-finish farms are mainly of the industrial type and represent the most common type of pig farms in Bulgaria. Thirty-two farrow-to-finish farms were recruited into the study from 11 districts. The districts chosen are the areas where industrial farms are present. Farms were randomly selected from the National Register of pig establishments in Bulgaria in 2020 (Table 1), but only some farmers were willing to participate in the study. From each district, 1 to 5 farms were visited, because in one district only one farrow-to-finish farm was present and it was sampled while in those districts with more farms of this type we choose to visit all (4–5) when possible. At the moment of the sampling in some districts a few farms were empty and, subsequently, not enrolled in this study. All the visited farms had a similar management. The population of finishers varied between the farms, but in general, farms with a capacity >1000 finishers dominated (22 farms), followed by farms with a capacity between 200 and 1000 finishers (6 farms), and small farms with <200 finishers (4 farms)

The farms were visited in the period between October 2020 and June 2021. In the BIOPIGEE project the sampling scheme of this study was planned with the aim to also detect *Salmonella* as well as for the detection of HEV.

Twenty pooled fecal samples per farm were obtained, with each consisting of 10 individual feces samples. Each individual sample, used to prepare the pooled sample, contained 10 g of fresh feces, collected preferably immediately after defecation with plastic gloves and a spoon, which were changed after each sample collection. In addition, it was recommended that fecal samples were best collected from as many different pens as possible from the targeted groups of pigs (see below).

This sample size provided sufficient sensitivity to detect at least one positive sample with 95% confidence even if the within-herd prevalence was as low as 2% and would estimate an expected within-farm prevalence of 10% with 15% precision [28,29,30].

The ratio of samples (finisher/gilt/dry sow) collected on farm was 50–40–10% (10 pooled fecal samples from finishers, 8 pooled fecal samples from gilts, and 2 pooled fecal samples from dry sows). In the context of the study design, the dry sows are pigs in the dry period (time interval from weaning to farrowing), gilts are female pigs of ~6 month of age before the first delivery of a litter of piglets, and finishers are pigs after weaning until they reach their market weight. In one of the tested farms in the Pazardzhik district, fatteners were not present at the sampling visit and so we sampled only dry sows and gilts.

The average slaughtering age of fattening pigs in Bulgaria is 4–6 months of age. The age of the fattening pigs sampled in our study were between 2.5 and 6 months. The sampled fattening pigs were split into two different age groups: finishers aged between 60 and 120 days (n = 197 pooled fecal samples) and finishers aged between 121 and 180 days (n = 123 pooled fecal samples). 

### 2.2. Virus Concentration and RNA Isolation

Samples were transported to the laboratory in cooling boxes so that the temperature did not exceed 8 °C. After arrival at the laboratory the obtained samples were either tested immediately or were stored at −20 °C. One gram of pooled samples was transferred to a 50 mL centrifuge tube and suspended in 9 mL of PBS. The suspension was vortexed for 60 s and centrifuged at 3000× *g* for 15 min. The clarified supernatants (100 µL) were immediately used for nucleic acid isolation. In order to determine the extraction recovery rate before RNA extraction, 10 µL Mengovirus was spiked in the fecal supernatants and in 100 µL PBS as a positive extraction control (PEC). The QIAmp^®^ Viral RNA MiniSpin Kit (Qiagen, Germany) was used for nucleic extractions following the manufacturer’s instructions.

The final elution was performed twice with 50 µL elution buffer, resulting in 100 µL nucleic acid extract. The nucleic acid extract was assayed immediately or stored at −80 °C.

### 2.3. HEV Specific One Step Real-Time RT-PCR

HEV RNA was detected using qualitative (presence/absence) one-step real-time RT-PCR.

The real-time RT-PCR reactions were carried out on an AriaDX Real-Time PCR System (Agilent, Santa Clara, CA, USA). For HEV RNA detection, we used oligonucleotide primers and a fluorescent probe as described previously [31]. The QuantiTechProbe RT-PCR Kit (Qiagen, Hilden, Germany) was used with 5 µL of RNA to prepare a reaction with a total volume of 25 µL. The cycling parameters were as follows: reverse transcription at 50 °C for 30 min, initial denaturation at 95 °C for 15 min, followed by 45 cycles of denaturation at 94 °C for 10 s, annealing at 55 °C for 20 s, and elongation at 72 °C for 1 min.

Each set of viral extracts contained a negative extraction control, a water control, and a positive target RNA control for each PCR run [32].

The extraction recovery was calculated by using Mengovirus [33]. For Mengovirus detection, we used oligonucleotide primers and a fluorescent probe described by Pinto et al. [17]. The QuantiTechProbe RT-PCR Kit (Qiagen, Hilden, Germany) was used with 5 µL of RNA in a final volume of 25 µL. The thermal conditions followed were reverse transcription at 55 °C for 60 min, initial denaturation at 95 °C for 5 min, followed by 45 cycles of denaturation at 95 °C for 15 s, annealing at 60 °C for 1 min, and elongation at 65 °C for 1 min [31,33,34]. The Mengovirus recovery rate (%) was calculated by comparing the *Ct* values of Mengovirus recovered from spiked fecal supernatants with the *Ct* value obtained from RNA derived by using Mengovirus (PEC) spiked in water, using the formula 2^−ΔCt^ × 100 [35]. Samples showing extraction recovery lower than 1% were considered not acceptable as suggested by standardized methods for food-borne virus detection [33].

### 2.4. Limit of Detection (LOD)

The 1st WHO International Standard for Hepatitis E virus RNA (PEI code 6329/10; Paul-Ehrlich Institute, Berlin, Germany) was used for the evaluation of the limit of detection (LOD). HEV-negative pig stool samples (in triplicates) were spiked with 2-fold dilutions ranging from 70.100 GE to 1.095 GE. This corresponds to 5.39 log_10_ copies/mL according to Baylis et al. [36]. The samples were subjected to RNA extraction following the protocol describe above and HEV RNA detection using HEV-specific real-time RT-PCR. The LOD was the dilution at which all three replicates showed a positive HEV RT-PCR result. 

The limit of detection (LOD) for HEV RNA was determined at 10.96 log_10_ copies/mL. 

### 2.5. Statistical Analysis

Statistical analysis was performed using Excel 2007 (Microsoft, Redmond, WA, USA). Chi-square tests were performed to detect differences between categories of variables and the HEV results. When the result was *p* < 0.05 this was determined to be statistically significant.

## 3. Results

The mean virus extraction recovery rate was 47.64% with a standard error of 1.26%. The samples (186) which tested negative for HEV RNA and had a recovery rate lower than 47.46% were re-tested by diluting the RNA 1:5. After re-testing the absence of HEV RNA was confirmed again.

Overall, HEV RNA was detected in at least one pooled fecal sample on 37.5% of the tested farms in Bulgaria (12/32).

The HEV RNA was most commonly (40.9%, 9/22) detected in farrow-to-finish farms housing more than 1000 fattening pigs, whereas only 30% (3/10) of the farms with 1000 fattening pigs or less had HEV RNA detected. However, this difference was not statistically significant (P_chi-sq_ > 0.05).

Furthermore, HEV RNA detection levels varied widely across administrative districts with a higher burden of HEV-positive farms in the eastern part of the country. In the Shumen and Gabrovo districts all the three tested farms were positive for HEV RNA (100%), while in the Pazardzhik, Vratsa, Montana, and Lovech districts it was not detected in any of the tested farms. Four farrow-to-finish farms in each of Stara Zagora and Yambol districts were surveyed and the results were similar, with HEV detected in two out of four farms in each district. A high proportion of farms were also HEV-positive in the Razgrad and Varna districts, 75% (3/4) and in 33.3% (1/3), respectively. In the Ruse district, only one out of a total of five farms gave a positive result for the presence of HEV RNA (Table 2). The number of farms in each region was too small to make useful statistical comparisons.

The overall percentage of HEV-RNA-positive pooled fecal samples was 10.8% (68/630, Table 3). Figure 1 shows the percentage of HEV-positive farms in the different administrative districts in Bulgaria. Sampled farms located in the western part of Bulgaria were free from the presence of the HEV. In the eastern and northeastern part a higher proportion of samples were HEV positive, with up to 100% tested farms positive for the presence of HEV. 

The occurrence of HEV RNA in pooled fecal samples collected from pigs belonging to different age categories in the farrow-to-finish pig farms is shown in Table 3.

HEV RNA was detected most frequently in pooled fecal samples from fattening pigs at 20.6% (66/320), followed by dry sows at 1.6% (1/62), and gilts at 0.4% (1/248). There was no significant difference between the results from dry sows and gilts, but the fatteners had significantly greater odds of being positive (P_chi-sq_ < 0.001) than the sows and gilts. Overall, the presence of HEV in pooled fecal samples in fatteners (20.6%) was higher than in breeding pigs (0.64%). HEV RNA was significantly (P_chi-sq_ = 0.04) more likely to be detected in pooled fecal samples from finishers aged between 60 and 120 days, 24.4% (48/197), rather than in older finishers, aged between 121 and 180 days (14.6%).

## 4. Discussion

The zoonotic nature of HEV transmission in European countries has become an emerging issue, especially in transplanted and immunocompromised patients [5].

The present study reported that HEV was frequently detected in farrow-to-finish pig farms in Bulgaria (12/32). The highest percentage of HEV-positive pooled fecal samples was from younger fatteners (24.4%). The fattening pig population aged between 4 and 6 months (shortly before being slaughtered) also showed a high positivity of 14.6%. The result underlines the potential risk of HEV entering into the food chain, representing a risk if pork food is not properly cooked. 

Similar studies have been carried out in many other European countries and the decrease in HEV positivity with age has also been reported. In Italy, a study conducted on six farms, found a higher HEV prevalence (42.2%, 27/64) in pigs aged 90–120 days than in pigs up to 120 days old (27%) [37]. In the Netherlands and Belgium, fecal samples from fatteners of 5–6 months of age were collected at slaughterhouses and were analyzed for HEV RNA. In the Netherlands, 15% of pigs were detected as HEV positive and 7% in Belgium [38], confirming similar findings with our results in older pigs. Results from Sweden were similar to our study, but they tested individual samples rather than pooled fecal samples [39]. The occurrence of HEV in pig farms in Eastern Europe was also investigated. A study conducted on five farms in Eastern Romania showed a HEV prevalence of 31% in fecal samples (6/19) from fattening pigs of between 2 and 4 months of age [40]. In Slovenia, 24.2% pooled fecal samples from fattening pigs (>10 weeks old) housed on seven farms tested HEV positive [41]. This latter result is comparable to our result for the percentage of positivity and for the use of pooled fecal samples for testing.

The aforementioned studies reported a higher percentage of HEV positive in younger animals. This could be explained by the susceptibility of pigs to HEV infections after the loss of maternal immunity, clearance of infection, production of anti-HEV IgG antibodies, and higher protection in older pigs that could explain the decrease in prevalence with age. 

This study has some limits, as some areas of the country were not sampled and so it is not representative of the whole country, but the main areas for industrial farms were tested. Furthermore, a limitation of the study is the absence of the HEV genotyping of positive samples, which would have led to useful information on subtype diversity in the sampled herds and regions. Unfortunately, this was not the original object of the project. Nevertheless, it is the first national study in Bulgaria carried out to detect HEV RNA in pig feces from different regions and age groups on farrow-to-finish farms. Studies on the occurrence of HEV in pigs in Bulgaria have been carried out since 2017 but mainly reported seroprevalence [26,27,42]. The conducted studies showed a high seroprevalence of HEV in the pig population in Bulgaria, ranging between 40% and 60% [21,24,26,27,43,44]. Comparing our data with previous studies is difficult because seroprevalence does not necessarily correspond to HEV shedding in feces. One study in Bulgaria reported molecular screening using qRT-PCR on 39 serum samples and HEV RNA was detected in 28.2% [43].

In our study, in the Ruse district (northeast of Bulgaria), only one out of five investigated farms was HEV RNA positive (20%). Previous data on IgG anti-HEV in pigs in this administrative district showed between 80% and 96.7% seroprevalence in finishers and 23% in sows in the two farms studied from the Ruse district [42]. The situation is similar to previous data on the presence of anti-HEV IgG in the Varna district, where the authors reported almost 50% seroprevalence in 2018 and 100% in 2019 for slaughter age pigs (6 months old) [42]. In our study, the presence of HEV RNA was also confirmed but in only one of the three studied farms in the same district of Varna. In the Shumen district the same authors reported a HEV seroprevalence of 53.3% and 100% in 2018 and 2019, respectively, in the two investigated farms, and 100% among pigs of slaughter age [42]. In our study, in the Shumen district, 20% of the finishers (4 months age) in both tested farms were HEV positive. This result confirms that seroprevalence is generally higher than the percentage of HEV RNA detected, which corresponds to the shedding period and which has a shorter duration compared to the duration of IgG [20]. This result could be explained since IgG anti-HEV reflects past HEV infection that does not necessarily correspond to the presence of HEV RNA. A higher value of seroprevalence is expected since the seroprevalence increases with the age of pigs, being up to 100% in adults [8,20].

The observed differences with the reported results from previous studies conducted in Bulgaria can be explained given the dynamics of HEV infection. Under natural conditions the acquisition of natural passive immunity through the colostrum ingested from the mother, gradual reduction of these passive antibodies at 2–2.5 months of age, and then seroconversion between 3.5 and 4 months of age usually corresponds with the peak of infection that is observed around 4 months of age. Pavio et al. [45] reviewed the fecal excretion of HEV, which showed a peak at 4.5 months of age (86% of pigs with fecal shedding of virus) and then gradually decreased to 41% at slaughter age (around 6 months). These data are also confirmed by Nakai et al. [46] who found that peak HEV fecal excretion was between 1 and 3 months of age (75 to 100% of animals), which then decreased to 7% of pigs at 5–6 months of age, which was in agreement with the finding from our study.

By mapping the presence of HEV in the different districts (Figure 1), no positive farms were detected in the western part. It is a characteristic of the eastern part of Bulgaria that a large number of pig farms with a capacity of >1000 finishers are predominant. Some risk factors for the introduction of HEV on farms have been presented in previous studies [47,48]. The poor hygiene conditions within the farm and in feed, water, and bedding were considered possible risk factors for the introduction of the virus in farms [47,48]. Additionally, outbreaks of African Swine Fewer in 2019, mainly in Eastern Bulgaria, caused the farm population to be replaced [49]. These factors could have contributed to the introduction of either suckling pigs still susceptible to the infection or HEV-positive pigs in farms and differences in results between the eastern and the western areas.

A limitation of the present study could be the use of pooled feces instead of individual fecal samples. However, the use of pooled feces provided robust results as described in a previous study on HEV detection in pigs. The authors assessed pooled sample sensitivity in artificial pig fecal pools and reported a high sensitivity up to a fifty-fold dilution for positive fecal samples, allowing for the detection of the virus even if a pen contained 50 animals, 49 of which were negative at the moment of sampling [50]. In our study, pooling provides the advantage of reducing the number of samples to be analyzed. The investigated farms were representative of industrial Bulgarian farms and so the results give a valuable indication of the presence of the pathogen.

## 5. Conclusions

In conclusion, domestic pigs represent an important zoonotic reservoir for HEV infection. Our findings confirm that HEV circulates in industrial pig farms in Bulgaria and, therefore, pigs are likely to be an important vector of the virus. In our study, we found pooled fecal samples that were positive for HEV RNA mainly in finishers, shortly before their transport to the slaughterhouse. This suggests that there are concerns about the potential transmission of this virus, through contact with pigs either during handling or through the consumption of contaminated pork products. To avoid possible circulation of HEV through pork production containment measures should be considered. The consumption of safe food products remains one of the most important commitments of any public policy worldwide. Assessing the risk factors associated with the introduction and circulation of HEV infection in pig farms remains a basic requirement for the prevention of pathogen transmission.

The obtained data would be useful for a risk assessment of pork food safety and would provide baseline data for establishing a system for the control and prevention of HEV infections in pigs, as well as for strategies to limit public health consequences.

## Figures and Tables

**Figure 1 pathogens-12-00673-f001:**
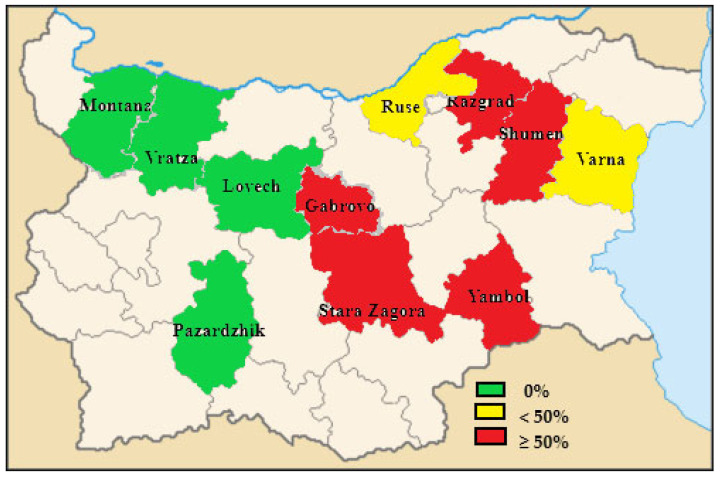
Percentage of HEV-positive pig farms in the different administrative districts in Bulgaria.

**Table 1 pathogens-12-00673-t001:** Numbers of tested farms by administrative districts.

Administrative District	Number of Sampled Farms	Total Pooled Samples per Districts
Montana	2	40
Vratsa	1	20
Lovech	2	40
Gabrovo	1	20
Ruse	5	100
Razgrad	4	80
Shumen	2	40
Varna	3	60
Pazardzhik	4	70 *
Stara Zagora	4	80
Yambol	4	80
In total	32	630

* In Pazardzhik district, only 10 samples were collected (from dry sows and gilts) from a farm, as no finishers were present at the moment of sampling.

**Table 2 pathogens-12-00673-t002:** Percentage of HEV-positive farms in different Bulgarian administrative districts.

Administrative District	Farms		
	Total Tested	Number of Positive Farms	% of Positive Farms
Razgrad	4	3	75
Pazardzhik	4	0	0
Vratza	1	0	0
Montana	2	0	0
Ruse	5	1	20
Lovech	2	0	0
Gabrovo	1	1	100
Stara Zagora	4	2	50
Yambol	4	2	50
Shumen	2	2	100
Varna	3	1	33.3
Total	32	12	37.5

**Table 3 pathogens-12-00673-t003:** Molecular detection of HEV RNA in pooled fecal samples according to different age categories in pigs from Bulgaria.

Production Stage	Total Number of Tested Pooled Fecal Samples	Number of Positive Pooled Fecal Samples	% of Positve Pooled Fecal Samples
Dry sows	62	1	1.6
Gilts	248	1	0.4
Total Breeding	310	2	0.64
Fattening pigs (60–120 days)	197	48	24.4
Fattening pigs (121–180 days)	123	18	14.6
Total fattening	320	66	20.6
Total (fattening and breeding)	630	68	10.8

## Data Availability

Data are available from the corresponding author upon request.

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
