# Peer review of "Detection of HEV RNA Using One-Step Real-Time RT-PCR in Farrow-to-Finish Pig Farms in Bulgaria"

_pathogens, 2023, doi:10.3390/pathogens12050673_

Round 1

Reviewer 1 Report (New Reviewer)

Author Response

Response to Reviewer 1 comments

The authors would like to thank the reviewer for the interest in this manuscript and the valuable comments and suggestions for improvement. Please find our responde to each comment below.

Comments concerning the content of the manuscript:

Point 1: A main problem of the manuscript is the impeded readability due to partly stronger linguistic weaknesses. A language revision should be carried out.

Response 1: We check the manuscript and language revision was carried out by a native English-speaking co-authors.

Point 2: In line 30, the authors describe that the farms were randomly selected. In line 306, however, the authors state that the farms were not selected randomly and that bias was present. The authors should urgently provide more information on what pattern was used to select the farms. Furthermore, how do the authors explain that the number of farms studied per district varies between 1 and 5? Especially the data on the Pazardzhik district, when compared to the existing literature, leave doubts about the representativeness of the selected farms.

Response 2: Farms were randomly selected among farrow-to-finish farmsq which are the most common type of farms in Bulgaria. We selected districts where pig farms are commonly concentrated. The farms were selected from National Register of pig establishments in Bulgaria in 2020, but some farmers did not allow visiting their farm. At the moment of the sampling in some districts a few farms were empty and subsequently not enrolled in this study.

The number of studied farms per district varied between 1 and 5 because in one district only one farrow-to-finish farm was present and it was sampledq while in those districts with more farms of this type we choose to visit all (4-5)q when possible.

Point 3. The number of HEV-positive farms varied from 0 to 100% between the districts. Do the authors have an explanation for the higher burden in the eastern part of the country?

Response 3. We do not know exactly, but it is a characteristic of the Eastern part of Bulgaria that a large number of pig farms with a capacity >1000 finishers dominate, as well due to the outbreaks of ASF in 2019, mainly in Eastern Bulgaria, the farm population was replaced.

Point 4. Starting at line 321, the authors describe the need to identify the risk factors of HEV introduction in pig farms. Are there any concrete proposals or solutions available?

Response 4. As few former studies have presented some risk factors for HEV introduction (Walachowski et al., 2014), but the reply is not available. The lack of hygienic conditions whithin the farm, in feed, water and bedding were considered possible risk factors for the introduction of the virus in farms (Walachowski et al., 2014; Galipo et al., 2023).

Point 5. Do the authors see an explanation for the fact that younger fattening pigs showed a markedly higher HEV load than older fattening pigs?

Response 5. We discussed in the text that a higher percentage of HEV positive animals among younger fattening pigs could be linked to higher suscptibility at this age, when the risk of infection for HEV is higher (due to lost of maternal immunity). The clearance of the infection probably occurred in older pigs that indeed resulted less frequently positive for HEV

Point 6. Do the authors have data on whether there is a correlation between the number of positive stool samples and the clinical condition of the pigs? For example, do the farms with detected HEV circulation show limitations in growth or higher mortality of the animals?

Response 6. We cannot reply to the question. Pigs did not show any symptoms when were sampled. We did not collect any data on the clinical condition of the pigs.

Point 7. The classification in dry sow, gilt and finisher should be explained to the reader.

Response 7. Thank you for this hint. We added an explanation the classification for dry sows, gilt and finishers in the M&M section: “In the context of the study design, the dry sows are pigs in the dry period (time interval from weaning to farrowing), gilts are female pigs of ~6 month of age before the first delivery of litter of piglets and finishers included pigs after weaning until they reach their market weight”.

Comments concerning the presentation of the manuscript:

Point 8. Please ensure consistent hyphenation, e.g. HEV-RNA/HEV RNA (e.g. lines 2, 15, 19 and 20) or RT-PCR/RT PCR (e.g. lines 2, 15 and 18).

Response 8. Thanks. We corrected.

Point 9. Affiliation 4 is listed, but it is not assigned to any author.

Response 9. We appologies. We corrected.

Point 10. Typo in line 31: „stand“

Response 10. We corrected

Point 11. Why does the citation start with reference 16 and reference 1 is in line 239? Please number in order of appearance in the text, not alphabetically.

Response 11. Thank you for this comment. We have corrected the citation as required by the journal.

Point 12. Please provide a reference in line 50.

Response 12. Corrected

Point 13. The presentation of the tables should be revised. For example, the numbers in Table 1 should be centered under the labels as are the districts. Also in table 2. The right four columns are not centered.

Response 13. We formatetted Table 1 and Table 2

Point 14. Please use consistent spellings for districts (for example, Pazardzhik line 279 and Pazardjick Table 2).

Response 14. Thank You for the notes, we corrected Pazardjick with the right spelling of the district – Pazardzhik.

Reviewer 2 Report (New Reviewer)

Hepatitis E virus (HEV) has recently become a significant public health issue. In this article, the authors have determined the frequency of HEV infection in pig farms in different regions of Bulgaria using real-time RT PCR. The study is well performed and holds high significance. 

Author Response

The authors would like to thank the reviewer for the interest in this manuscript  and the positive evaluation.

Reviewer 3 Report (New Reviewer)

Although the article presents interesting results that contribute to the local epidemiology in Bulgaria, it is not written in a clear and concise manner, and the data presented is rather poor. The first question that arises is: why was the genotype not determined (with previous amplification by conventional PCR in those positive samples)? This would have enriched the work, providing valuable information to the molecular epidemiology of this country.

Furthermore, it is not clear whether individual samples or only pools were tested. For example: in the Results section, the authors state: “The overall prevalence of HEV RNA in the pooled faecal samples was 10.8% (68/630)”. How do you know? Did you tested single samples when a pool resulted positive? Or did you test all the samples individually? In this case, why were pools made? This is not clear in the mentioned sentence and the following ones. This should explained in the text.

Table 3: Again: how do you know the exact number of HEV positive samples if you tested pools? Please clarify above in the text.

Minor comments:

Line 70: please, remove “but it is unknown if it is a zoonoses”, so that the sentence finishes in the word genus.

Line 191: change “hepatitis E virus” by “HEV”.

Conclusion:

-Lines 314: please, change “and therefore they are likely to be an important vector of the virus, both in the pig population and in humans” by “and therefore these animals are likely to be an important reservoir of the virus, transmitting it within the pig population and also to humans.”

-Line 322: please, remove the word “infection”.

Author Response

Response to Reviewer 3 comments

The authors would like to thank the reviewer for the interest in this manuscript and thoughtful and for the valuable comments and suggestions for improvement. Please find our response to each comment below.

Point 1: Why was the genotype not determined (with previous amplification by conventional PCR in those positive samples)? This would have enriched the work, providing valuable information to the molecular epidemiology of this country

Response 1: The authors thanks for your interest about the circulating genotypes. We understand that this information would provide valuable information on HEV epidemyology in Bulgaria, but the genotyping HEV was not object of this work.

We actually have resent information about genotypes circulating in Bulgarian farms, but this is a scope of another survey and the data will be published soon.

Point 2: it is not clear whether individual samples or only pools were tested. For example: in the Results section, the authors state: “The overall prevalence of HEV RNA in the pooled faecal samples was 10.8% (68/630)”. How do you know? Did you tested single samples when a pool resulted positive? Or did you test all the samples individually? In this case, why were pools made? This is not clear in the mentioned sentence and the following ones. This should explained in the text.

Response 2: We only tested pooled faecal samples. We improperly used the term “prevalence” that was revised over the whole manuscript. Since we followed the same sampling scheme in all farms, collecting the same number of faeces that were used to make the pools and the same categories of pigs we used the percetage of positive pool fecal samples for comparisons among farms. We explained how the pools were build in section M&M: pooled samples (10 individual pinches of faeces per pooled sample) and this sample size provided sufficient sensitivity to detect at least one positive sample with 95% confidence even if the within-herd prevalence was as low as 2% and would estimate an expected within-farm prevalence of 10% with 15% precision.

Point 3. How do you know the exact number of HEV positive samples if you tested pools? Please clarify above in the text

Response 3. We tested only pooled samples. Please see the previous reply.

Minor comments:

Point 4. Line 70: please, remove “but it is unknown if it is a zoonoses”, sothat the sentence finishes in the word genus.

Response 4. Corrected

Point 5. Line 191: change “hepatitis E virus” by “HEV”.

Response 5. Corrected

Point 6. Lines 314: please, change “and therefore they are likely to bean important vector of the virus, both in the pig population and in humans” by “and therefore these animals are likely to be animportant reservoir of the virus, transmitting it within the pig population and also to humans.”

Response 6. We corrected

Point 7. Line 322: please, remove the word “infection”.

Response 7. We corrected

Round 2

Reviewer 3 Report (New Reviewer)

I still find the article unclear, especially in the methodological part. It is not explicit how many individual samples are used to perform the stool pools, how many pools are analyzed and what is meant by “overall percentage of 10.8% (68/630)”. Do you mean that 630 pools were analyzed? Or 630 individual samples grouped in pools? A figure with the number of initial samples and the number of pools performed would help to understand this.

The authors refer to the Table 2 to explain this value [“overall percentage of 10.8% (68/630)”] in line 193. However, Table 2 does not have the positive or negatives values. It seems this Table lost the format, may be when it was saved as a pdf file. Please check this. Or may be the authors wanted to refer to Table 3.

Table 3 is confusing. Please, eliminate line which states “Total fattening”, so that the line that says "Total" reflects the sum of all the columns.

In the Discussion section, it should be included another limitation of the study, which is not having carried out the HEV genotyping. This is easy to perform, and is a major limitation of the study.  

Author Response

The authors would like to thank the reviewer for the interest in this manuscript and thoughtful and for the new comments and suggestions for improvement. Please find our response to each comment below.

Point 1: I still find the article unclear, especially in the methodological part. It is not explicit how many individual samples are used to perform the stool pools, how many pools are analyzed and what is meant by “overall percentage of 10.8% (68/630)”. Do you mean that 630 pools were analyzed? Or 630 individual samples grouped in pools? A figure with the number of initial samples and the number of pools performed would help to understand this

Response 1: We tested only pooled faecal samples. We have made changes in the sentence (lines 105-106) and explained twenty pooled faecal samples per farm were obtained, with each consisting of 10 individual faeces samples. Each individual sample, used to prepare the pooled sample, contained 10 g of fresh faeces, collected preferably immediately after defecation. In lines 109-111 we clarify that it was recommended for the faecal samples to be collected from as many different pens as possible from the targeted groups of pigs. The overall percentage of 10.8% refers to analyzed of 630 pooled samples (line 109: of HEV RNA positive pooled faecal samples).

Point 2: The authors refer to the Table 2 to explain this value [“overall percentage of 10.8% (68/630)”] in line 193. However, Table 2 does not have the positive or negatives values. It seems this Table lost the format, may be when it was saved as a pdf file. Please check this. Or may be the authors wanted to refer to Table 3.

Response 2: With regards to  the revision comment which we recieved in first round, we reformated table 2 and excluded the data about pooled samples. We thank you for highlighting this problem, as we can now see that the citing of Table 2 is not realy correct. We have redacted line 193, where we have added a reference to Table 3. In line 203-204 we have slighlty changed it because figure 1 now only refers to farms, and not to the pool.

Point 3. Table 3 is confusing. Please, eliminate line which states “Total fattening”, so that the line that says "Total" reflects the sum of all the columns.

Response 3. We thank you for this comment, but we think that this row is useful and has not been removed. To make the information in Table 3 clearer, we have added additional row with the total breeders and the last row is redacted to “Total (fattening and breeders)”. This now provides clearer distinction of the ‘subtotals’ for finishers and breeders, and then the overall total.

Point 4. In the Discussion section, it should be included another limitation of the study, which is not having carried out the HEV genotyping. This is easy to perform, and is a major limitation of the study.

Response 4. We have included this limitation of the study in Discussion section (lines 260-263).

Round 3

Reviewer 3 Report (New Reviewer)

I still have doubt with the number of samples. The authors satated that 20 pooled samples were taken from each farm. There were 32 farms included, so the total number of pooled samples was: 20x32= 640. But in the text, you mentioned 630 pooled samples. Can you explain this difference?

The rest of the corrections are OK.

Author Response

The authors would like to thank the reviewer for the new question.  Please find our response  below:

Point 1: I still have doubt with the number of samples. The authors satated that 20 pooled samples were taken from each farm. There were 32 farms included, so the total number of pooled samples was: 20x32= 640. But in the text, you mentioned 630 pooled samples. Can you explain this difference?

Responce 1: I would respond that the difference in this numbers is explained in lines 122-124, but to make this clearer we have made changes to Table 1 and added a footnote.

This manuscript is a resubmission of an earlier submission. The following is a list of the peer review reports and author responses from that submission.

Round 1

Reviewer 1 Report

The manuscript “Detection of HEV RNA by one-step real-time RT-PCR in far-2 row-to-finish pig farms in Bulgaria” reports the results of a prevalence study on HEV in pig farms in Bulgaria. The information are new and relevant, as few data are available for Eastern Europe countries.  However, some relevant issues (see general comments) should be address before considering publication.

General comments:

1) Sampling

- Sampling plan: the Authors state that farms were selected from 11 districts by random selection. The absence of stratification (by number of farms at national and district level) leads to questioning the representativeness of the results at the country level. The Authors should clarify if the sampling scheme ensure such a national representativeness or, if not, should clearly discuss this limitation.

- Samples pooling: it is stated (lines 218-220) that pooling of 10 individual samples ensure sufficient sensitivity to detect one positive sample if the within-herd prevalence is as low as 2% and would estimate an expected farm prevalence of 10% (with what precision level?). The statement is supported by a reference to an EFSA baseline survey on Salmonella in pigs.  It is not clear how the Authors established this level of sensitivity on the basis of the cited baseline survey, considering that pooling of samples affects sensitivity in different ways if the targets are subjected to analytical procedures including pre-enrichment stages or if they undergo direct detection (as in the case of HEV).

- Ratio of samples by pig type (lines 221-222): as for the farm selection, it is not clear if this ratio represents or not a distribution comparable to that of the pig farming system in Bulgaria or if it was an approximation established for sampling convenience. This aspect should be clarified and discussed.

2) Analytical approach

- The manuscript does not include essential information on the sensitivity of the detection method. What is the LOD expressed in genome equivalents per gram of pooled feces? What is the amount of pooled feces tested in each PCR reaction? A more detailed description of the extraction procedure (starting amount of sample, recovered volume after centrifucation, volume undergoing RNA extraction) should be included.

- The use of a positive extraction control (Mengovirus) is described. However, the extraction control was applied to the fecal supernatant and not on the pooled feces before dilution with PBS and clarification by centrifugation. In this way the positive control acts exclusively as a control for the nucleic acid extraction phase and not for the whole extraction process of the fecal samples (in which viral particle may be lost during the centrifugation phase or for attachment to the organic matter/particles). It is therefore not correct to use this positive control to assess the recovery rate, as it only covers a part of the procedure. Besides, the ‘permissive’ acceptability criterion of 1% (which is used for concentration/extraction controls in ISO 15216 due to the fact that the loss during concentration procedures is accounted for) was applied, despite the fact that the positive extraction control only accounts for nucleic acid extraction.  Finally, the extraction efficiency results are not reported or discussed in the manuscript. Was the average close to 100% as expected for a nucleic acid extraction? What was the range?

- There is no mention of the use of an inhibition control in the PCR. How was inhibition ruled out? Could Mengovirus results provide information on the presence of PCR inhibitors instead of on ‘recovery’? If no inhibition control was applied, a risk for underestimation of prevalence results may be present and should be addressed and discussed by the Authors.

- RT-qPCR acronym is used for quantitative PCR; as the test was used in a qualitative way, the ‘q’ should be deleted in the text

3) Statistical analysis:

A thorough revision of the results should be performed as the manuscript alternatively reports a total number of samples of either 640 or 630 (and associated percentages calculated with the two denominators). Based on table 1, one farm in Pazardjick did not provide 20 samples but only 10. This point should also be clarified in the sampling plan.

4) Manuscript organization: Given the current structure of manuscript published in Pathogens, the Materials and Methods section should be included between the Introduction and the Results sections.

5) Editing: please check the manuscript for English (phrasing, words misuse, verb tenses). Please also check full stops as they are missing in several instances.

Specific comments:

-          Introduction (lines 36-41): the different transmission routes of genotypes 1-2 (mostly waterborne in developing countries) and 3-4 (foodborne in industrialized countries) should be mentioned

-          Introduction (lines 42-46): also mention of risk for pregnant women (particularly in developing countries) should be included in the general description of the public health problems posed by HEV

-          Introduction (lines 47-60): line 47 – please, change “increasing popular” (relevant? acknowledged?); line 58 – ‘forestry officials’; lines 55-57 – please specify in the sentence that is an EFSA opinion summarizing the available information.  Please include in this section also a mention to the other reservoir species that may contribute to foodborne transmission (e.g. wild board, cervids).

-          Introduction (lines 79-81): The final sentence is not coherent with the reported data nor with the conclusions as no estimate of infected pig at slaughtering is discussed in the manuscript. Please revise.

-          Results: Please include, if available, information on recovery/inhibition. Avoid redundancy of expressions as “[number] out of a total of…” (number/number can be reported).  Lines 112-113: it is not clear were a 100% prevalence was detected in samples (100% can be seen in farms results but not in samples).

-          Table 1: please include a headline separating “Farms” form “Pooled samples” and, below, for each one, report the categories “total tested”, “positive”, “% positive”. Please also include a closing line with the totals. The information provided by this table could also be strengthened by including lines for each farm (anonymized), detailing the number/percentage of positive samples for each one. These data, which could be useful for future re-analysis could also be provided as supplementary.

-          Figure 1: a district not mentioned in the paper (Veliko Tarnovo ?) is also reported in the figure with a prevalence value >20%

-          Discussion: for a better comprehension of the significance of the results for public health, the Authors should clarify what is, on average, the slaughtering age in Bulgaria. In the manuscript it is reported that detection of HEV was achieved in finishers shortly before their transport to slaughterhouse, but it should be clarified if 4- to 6-months age pigs are considered a statistical group that “approximates” the slaughtering age.

-          Discussion: the difference between seroprevalence data and HEV RNA detection is reported several times in the discussion (lines 175-177, 186-188, 207-209).  This should be summarized in a single, comprehensive discussion.

-          Discussion (line 160): considering the European perspective of the scientific research, I would suggest to refer to these countries as “EU central and eastern countries”, according to EuroVoc.

-          Discussion (lines 166-168): this sentence is incomplete.

-          Discussion (lines 170-171): These results are not reported in the Results section. If a specific distribution of positive results was noticed for certain farms or district, it should be analytically reported in that section, before commenting/discussing it.

-          Discussion (lines 197-204): please clarify in the sentence that the reported information are related to a specific study, e.g. by writing “Pavio et al reviewed…” (i.e. fecal shedding of HEV is not present in 41% of tested animals in all conditions/countries/etc). Please also consider citing the original articles instead of a paper reviewing the data.   

Reviewer 2 Report

Comments and suggestions for authors

The manuscript “Detection of HEV RNA by one-step real-time RT-PCR in farrow-to-finish pig farms in Bulgaria” determined the presence of HEV RNA in industrial pig farms in several distinct regions in Bulgaria. This study provided data which can be beneficial in the establishment of policy for the control of HEV infection in industrial pig farms as well as for the prevention of the HEV transmission to human population, particularly in Bulgaria.  

However, I have concerns about the quality of the data presented in the manuscript. The data presented are too superficial and rather preliminary to fully support the authors’ conclusion that the positivity of HEV RNA in the fecal samples of the pigs in the investigated farms raised concerns about potential zoonotic transmission of HEV from pig to human (at least in Bulgaria). Consequently, the overall study is incomplete and does not strongly support the main conclusion. More convincing data are required to support the conclusion, for example, by providing the data on the genotypes circulating in the pig farms and to compare the identity with those circulating in human population in Bulgaria. The authors need to perform more in-depth investigation to substantiate their claims. 

Other major concerns are regarding the scientific flaws present in this manuscript. The manuscript is not presented in logical order, and therefore, is difficult to follow. The Discussion is not well organized and is fragmented. It lacks several important points where it should include not only comparison with other studies, but also the explanation behind the different/similar results with other studies, the significance or impact of the results of this study to human population in Bulgaria, the limitation of the present study, what future studies are warranted, etc. In addition, the authors must be more systematic and rigorous in comparing their observation with what is currently known. They should provide more detailed explanations throughout the manuscript. Overall, the data are poorly articulated, and the manuscript is difficult to read.

Of note, the authors stated that the HEV RNA-positive fecal samples were found in healthy (without clinical signs) finishers” (lines 24, 271). This sentence tends to mean that pig infected with HEV is generally symptomatic, and thus the findings of positive HEV RNA in the pig fecal samples are considered as new. Please note that HEV infection in pig is known to be asymptomatic, a knowledge that has been established in a large body of literature. 

Reviewer 3 Report

Authors collected the pooled samples from different pig farms and detected HEV RNA from those samples with one step PCR method. The result indicated that HEV-RNA positive rates among the tested farms were 36.3% and positive rates were different pigs at different age. In general, this result and method were not novel and study didn’t provide new information. Thus I suggest to reject.

Major problems

1.    All samples were pooled samples, didn’t detected individual samples for HEV RNA. Didn’t analyze HEV genotypes for positive samples.

2.    One conclusion is “In our study, we found positive pooled fecal samples for HEV RNA in healthy (without apparent clinical signs) finishers, shortly before their transport to the slaughter”. This conclusion should be confirmed by detecting HEV antibody and antigens from serum samples

3.    The positive rates were different at different age pigs. Didn’t mention whether all the samples were collected from same age pigs among different farms.